# Rv3723/LucA coordinates fatty acid and cholesterol uptake in *Mycobacterium tuberculosis*

**Evgeniya V Nazarova, Christine R Montague, Thuy La, Kaley M Wilburn, Neelima Sukumar, Wonsik Lee, Shannon Caldwell, David G Russell, Brian C VanderVen\***

Department of Microbiology and Immunology, College of Veterinary Medicine, Cornell University, Ithaca, United States

**Abstract** Pathogenic bacteria have evolved highly specialized systems to extract essential nutrients from their hosts. *Mycobacterium tuberculosis* (Mtb) scavenges lipids (cholesterol and fatty acids) to maintain infections in mammals but mechanisms and proteins responsible for the import of fatty acids in Mtb were previously unknown. Here, we identify and determine that the previously uncharacterized protein Rv3723/LucA, functions to integrate cholesterol and fatty acid uptake in Mtb. Rv3723/LucA interacts with subunits of the Mce1 and Mce4 complexes to coordinate the activities of these nutrient transporters by maintaining their stability. We also demonstrate that Mce1 functions as a fatty acid transporter in Mtb and determine that facilitating cholesterol and fatty acid import via Rv3723/LucA is required for full bacterial virulence in vivo. These data establish that fatty acid and cholesterol assimilation are inexorably linked in Mtb and reveals a key function for Rv3723/LucA in in coordinating thetransport of both these substrates.

\*For correspondence: bcv8@ cornell.edu

**Competing interests:** The authors declare that no competing interests exist.

## Introduction

*Mycobacterium tuberculosis* (Mtb), the causative agent of human tuberculosis (TB), is responsible for more than 1 million deaths each year and the bacterium currently infects nearly ~1.5 billion individuals. A hallmark of TB is that infected individuals rarely develop active TB disease and most infections (~90%) remain in a latent or asymptomatic state. Mtb is exquisitely adapted to survive within the human host and the bacterium's ability to metabolize host-derived lipids (fatty acids and cholesterol) is thought to enable bacterial persistence for long periods of time. Thus, elucidating the mechanisms involved in nutrient uptake and metabolism in Mtb will help us better understand important host-pathogen interactions and may identify new vulnerabilities for drug discovery.

Multiple lines of evidence indicate that host lipids (fatty acids and cholesterol) serve as critical carbon sources for Mtb during infection. In the 1950s, it was determined that Mtb propagated in mammalian tissues preferentially metabolizes fatty acids (*Bloch and Segal, 1956*) and numerous studies since have repeatedly demonstrated that lipid metabolism promotes Mtb survival during infection (*Marrero et al., 2010*; *McKinney et al., 2000*; *Muñoz-Elías et al., 2006*). Mtb also has the ability to metabolize cholesterol (*Van der Geize et al., 2007*) and the utilization of this nutrient is critical for Mtb survival within macrophages (*VanderVen et al., 2015*) and in various in vivo infection models (*Chang et al., 2007*; *Hu et al., 2010*; *Nesbitt et al., 2010*; *Pandey and Sassetti, 2008*; *Yam et al., 2009*).

Although Mtb's remarkable capacity to assimilate and metabolize lipids is considered a defining characteristic of this pathogen (*Cole et al., 1998*), the mechanisms underlying fatty acid uptake by Mtb have remained unknown. The mycolic acid-containing cell envelope of Mtb constitutes a unique

barrier for the import of hydrophobic molecules and likely explains why the Mtb genome does not encode canonical fatty acid transporters that are typically found in other bacterial systems (*Black et al., 1987*; *Theodoulou et al., 2016*; *van den Berg et al., 2004*). Instead, it is thought that Mtb uses Mce proteins to import hydrophobic nutrients across the bacterial cell envelope. The Mce family of proteins in Mtb were originally associated with mammalian cell entry (*Arruda et al., 1993*) and are classified as part of the MlaD superfamily (cl27420) based on the presence of one or more Mce domains (*Casali and Riley, 2007*). Proteins containing Mce domains have been implicated in the transport of hydrophobic molecules in various bacteria and chloroplasts (*Awai et al., 2006*; *Malinverni and Silhavy, 2009*; *Mohn et al., 2008*; *Roston et al., 2011*; *Sutterlin et al., 2016*; *Xu et al., 2008*). The aggregate data now indicate that Mce proteins mediate transport of hydrophobic molecules across double-membranous structures in cells (*Ekiert et al., 2017*; *Thong et al., 2016*). Mtb imports cholesterol across the mycobacterial cell envelope via the multi-subunit Mce transporter, termed Mce4 (*Pandey and Sassetti, 2008*) and the Mtb genome contains four total unlinked *mce* loci (*mce1-mce4*). Although the functions of proteins encoded in the *mce1-mce3* loci are unknown, the similarities shared across the *mce1-4* loci suggest that all these loci encode transporters responsible for the assimilation of hydrophobic substrates in Mtb.

Most of the genes required for the growth of Mtb on cholesterol as a sole carbon source have been mapped (*Griffin et al., 2011*) but, little is known about how Mtb utilizes cholesterol in the presence of other nutrients. Mtb can co-metabolize simple carbon substrates in vitro (*de Carvalho et al., 2010*), which suggests that the bacterium has the capacity to utilize fatty acids and cholesterol simultaneously in vivo. Mtb likely encounters both fatty acids and cholesterol during infection. For example, Mtb can induce foamy macrophage formation and can reside within these fatty acid and cholesterol-loaded cells (*Peyron et al., 2008*). It is also thought that Mtb is capable of persisting within the necrotic centers of human granulomas, an environment that is rich in fatty acids and cholesterol (*Kim et al., 2010*). Mtb also has the capacity to metabolically integrate the utilization of fatty acids and cholesterol when growing in macrophages where Mtb utilizes fatty acids to balance the incorporation of cholesterol-derived metabolic intermediates into biosynthesis of bacterial cell wall lipids (*Lee et al., 2013*).

Here, we performed a forward genetic screen to identify Mtb mutants defective in cholesterol utilization when the bacteria are grown in the presence of fatty acids. We identified Rv3723, a protein of unknown function and determined that this protein is required for the import of both fatty acids and cholesterol by Mtb during growth in macrophages and in axenic culture. Thus, we renamed this previously uncharacterized protein Rv3723, as LucA (lipid uptake coordinator A). We determined that LucA facilitates fatty acid and cholesterol uptake in Mtb by stabilizing protein subunits of the Mce1 and Mce4 transporters. These studies also establish that the Mce1 complex transports fatty acids into Mtb and that LucA is required for full virulence in vivo. Together, these findings demonstrate that fatty acid and cholesterol import in Mtb is integrated via LucA and that coordination of these activities is necessary to support Mtb pathogenesis.

## Results

### Identifying cholesterol utilization genes in Mtb

Using a forward genetic screen, we identified genes involved in cholesterol utilization when Mtb is grown in media containing a mixture of fatty acids and carbohydrates. To do this, we used the Mtb H37Rv mutant (Δ*icl1*::hyg) (*McKinney et al., 2000*) which has a synthetic lethal phenotype and fails to grow in rich media containing cholesterol. This growth inhibition is linked to the accumulation of one or more toxic cholesterol-derived intermediates produced by the methylcitrate cycle, which accumulate in the bacteria when *icl1* is nonfunctional (*Eoh and Rhee, 2014*; *Lee et al., 2013*; *VanderVen et al., 2015*). We reasoned that mutations in the cholesterol utilization pathway would rescue the growth inhibition of Δ*icl1*::hyg by preventing the formation of the cholesterol-dependent toxic methylcitrate cycle intermediate(s) (*Figure 1—figure supplement 1A*). Therefore, we isolated transposon rescue mutants in a Δ*icl1*::hyg background that gained the ability to grow on media containing cholesterol. In total, 133 clones were isolated and the mutations were mapped to 19 separate genes (*Figure 1—figure supplement 1B*). The rescue phenotype was further confirmed for 16 of the 19 mutants in liquid media containing cholesterol (*Figure 1—figure supplement 1B*).

Nine of the 16 identified genes had been predicted to be required for growth of Mtb on cholesterol as a sole carbon source (*Griffin et al., 2011*). These include *rv1130/prpD* and *rv1131/prpC*, which encode enzymes of the methylcitrate cycle. Inactivating *rv1130/prpD* and *rv1131/prpC* most likely limits production of the toxic methylcitrate cycle intermediate(s) that accumulate in the absence of *icl1* and this allows growth of these mutants (*Eoh and Rhee, 2014*; *VanderVen et al., 2015*). This screen also identified mutations in genes encoding cholesterol catabolic enzymes *rv3545c/cyp125*, *rv3568c/hsaC*, and *rv3570c/hsaA* (*Capyk et al., 2009*; *Dresen et al., 2010*; *Yam et al., 2009*). These mutations likely rescue growth by reducing the amount of propionyl-CoA released from cholesterol that can feed into the methylcitrate cycle. These findings are consistent with the observation that chemically inhibiting the PrpC or the HsaAB enzymes is sufficient to rescue growth of the *Δicl1*::hyg mutant in cholesterol media (*VanderVen et al., 2015*).

### *lucA* encodes a membrane protein involved in cholesterol metabolism

Seventeen sibling mutants carrying Himar transposon insertions in the *lucA* gene (*Δicl1*::hyg, *lucA*::TnHimar) were identified with the screen (*Figure 1—figure supplement 1B*), suggesting that *lucA* is required for cholesterol utilization by Mtb. Orthologs of *lucA* are restricted to *Mycobacterium* spp., and this gene encodes an uncharacterized putative integral membrane protein. Additionally, previous genetic epistasis studies implicated *lucA* in cholesterol utilization in vivo (*Joshi et al., 2006*), but, the function of this gene/protein has not yet been described. Using genetic complementation, we confirmed that the transposon insertion in the *lucA* gene is responsible for the growth rescue of

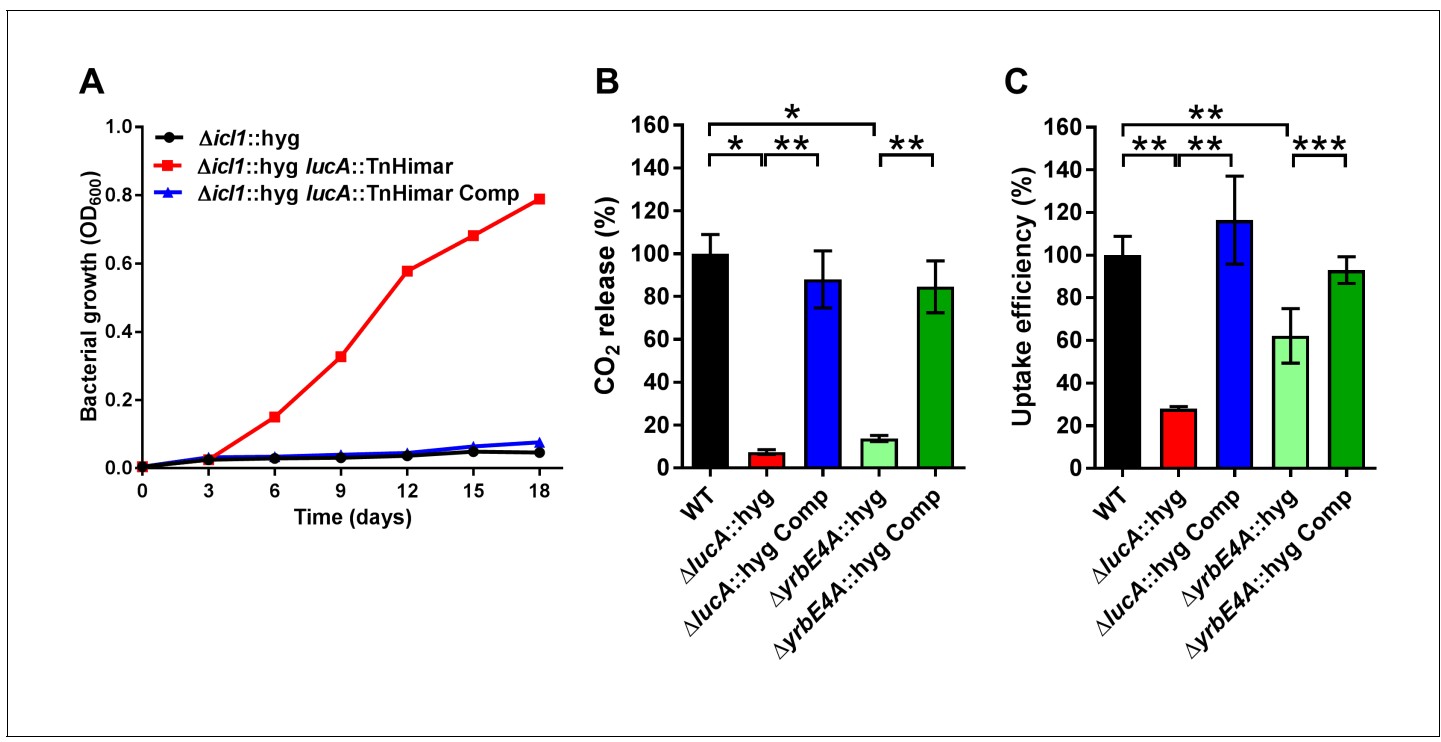

**Figure 1.** LucA facilitates cholesterol uptake in Mtb. (**A**) Complementation with *lucA* restores cholesterol toxicity in the *Δicl1*::hyg *lucA*::TnHimar mutant. (**B**) The *ΔlucA*::hyg and *ΔyrbE4A*::hyg mutants are defective in the catabolic release of $^{14}CO_2$ from [4-$^{14}$C]-cholesterol. (**C**) The *ΔlucA*::hyg and *ΔyrbE4A*::hyg mutants are defective in cholesterol uptake. Representative $^{14}$C-cholesterol uptake data used in these calculations (*Figure 1—figure supplement 2C,D*). Data are means ± SD (n = 4). *p<0.0005, **p<0.005, ***p<0.05 (Student's t test).

The following figure supplements are available for figure 1:

**Figure supplement 1.** Rational for genetic screen and summary of mutants selected by the screen.

**Figure supplement 2.** Mutant construction and kinetic analysis of cholesterol uptake.

the $\Delta icl1$::hyg, lucA::TnHimar mutant in cholesterol-containing media (**Figure 1A**). To ascertain subcellular localization, the LucA protein was fused to green fluorescent protein (LucA-GFP) and expressed in a wild-type strain of Mtb that constitutively expresses a cytosolic mCherry fluorescent protein. Confocal microscopy revealed a peripheral distribution for LucA-GFP, which is consistent with LucA having a cell membrane or cell envelope localization (**Figure 1—figure supplement 1C**).

## LucA facilitates cholesterol uptake and metabolism

Given the potential role of LucA in cholesterol metabolism, we constructed lucA mutant in Mtb Erdman by allelic exchange (Mtb $\Delta lucA$::hyg) (**Figure 1—figure supplement 2A**). We first quantified cholesterol metabolism in the $\Delta lucA$::hyg mutant by measuring the catabolic release of $^{14}CO_2$ from [4-$^{14}$C]-cholesterol (**VanderVen et al., 2015**). This assay detected a 90% reduction in the amount of $^{14}CO_2$ released from [4-$^{14}$C]-cholesterol in the $\Delta lucA$::hyg mutant relative to the wild-type and the complemented strains (**Figure 1B**). The reduction in $^{14}CO_2$ release was not due to a growth defect of the mutant (**Figure 1—figure supplement 2B**), and the percent $^{14}CO_2$ released was normalized to bacterial biomass.

To determine if lucA is required for cholesterol uptake in Mtb, we next quantified the rate of [4-$^{14}$C]-cholesterol uptake using an established assay (**Pandey and Sassetti, 2008**). We found that the $\Delta lucA$::hyg mutant assimilates 70% less cholesterol relative to the wild-type and complemented controls (**Figure 1C**). The cholesterol uptake rates were derived from the bacterial associated radioactive counts (**Figure 1—figure supplement 2C**) and the cholesterol uptake rate in wild type was set to 100% and the rates for the other strains were expressed as a ratio relative to wild type. The levels of cholesterol metabolism and uptake in the $\Delta lucA$::hyg strain was comparable to a Mtb Erdman mutant lacking the permease subunit of the Mce4 cholesterol transporter Rv3501c/YrbE4A ($\Delta yrbE4A$::hyg) (**Figure 1B,C** and **Figure 1—figure supplement 2D**). The residual cholesterol that is imported in the $\Delta yrbE4A$::hyg mutant may be transported by an unknown alternative cholesterol transporter in Mtb. Together these data demonstrate that lucA is required for the uptake of cholesterol and the subsequent metabolism of the sterol.

## The $\Delta lucA$::hyg mutant is defective in cholesterol utilization during infection in macrophages

We next analyzed the transcriptional profile of the $\Delta lucA$::hyg mutant during macrophage infection. These experiments revealed that the KstR regulon was strongly down-regulated in the $\Delta lucA$::hyg mutant (**Figure 2**). The curated gene expression data is compiled in **Figure 2—source data 1**. The KstR regulon is controlled by the transcriptional repressor, KstR which regulates expression of the genes encoding enzymes responsible for metabolizing the side chain and A/B rings of cholesterol (**Kendall et al., 2007**). The KstR regulon is activated in a 'feed forward' manner when KstR is de-repressed by binding to the second cholesterol degradation intermediate, 3-oxocholest-4-en-26-oyl-CoA (3OCh-CoA) (**Ho et al., 2016**). Thus, down-regulation of the KstR regulon in the $\Delta lucA$::hyg mutant is consistent with a decrease in cholesterol uptake by the $\Delta lucA$::hyg mutant and suggests that 3OCh-CoA may not be produced to levels that are sufficient to de-repress the KstR regulon in the mutant during infection in macrophages.

Mtb assimilates cholesterol-derived propionyl-CoA into central metabolism via the methyl-malonyl pathway and the methylcitrate cycle and the genes encoding enzymes of these pathways are normally highly expressed when Mtb metabolizes cholesterol (**Griffin et al., 2012**; **Savvi et al., 2008**). We found that expression of the methyl-malonyl and the methylcitrate cycle genes were also down-regulated in the Mtb $\Delta lucA$::hyg mutant (**Figure 2**). In Mtb propionyl-CoA, activates the rv1130/prpD promoter to enhance the bacterium's ability to assimilate cholesterol-derived metabolic intermediates via the methyl-citrate pathway (**Griffin et al., 2012**; **Masiewicz et al., 2012**). Thus, to monitor cholesterol breakdown in Mtb during infection in macrophages, we constructed a reporter vector where the rv1130/prpD promoter controls GFP expression while mCherry is constitutively expressed (prpD'::GFP smyc'::mCherry). We validated this reporter strain and determined by flow cytometry that GFP fluorescence increased >20 fold in wild-type Mtb when the bacteria were grown in media containing cholesterol or propionate (**Figure 2—figure supplement 1A,B**). Using this reporter, we observed a ~75% reduction in GFP fluorescence in the $\Delta lucA$::hyg strain relative to the wild type and complemented strain controls when the bacteria were grown in media containing

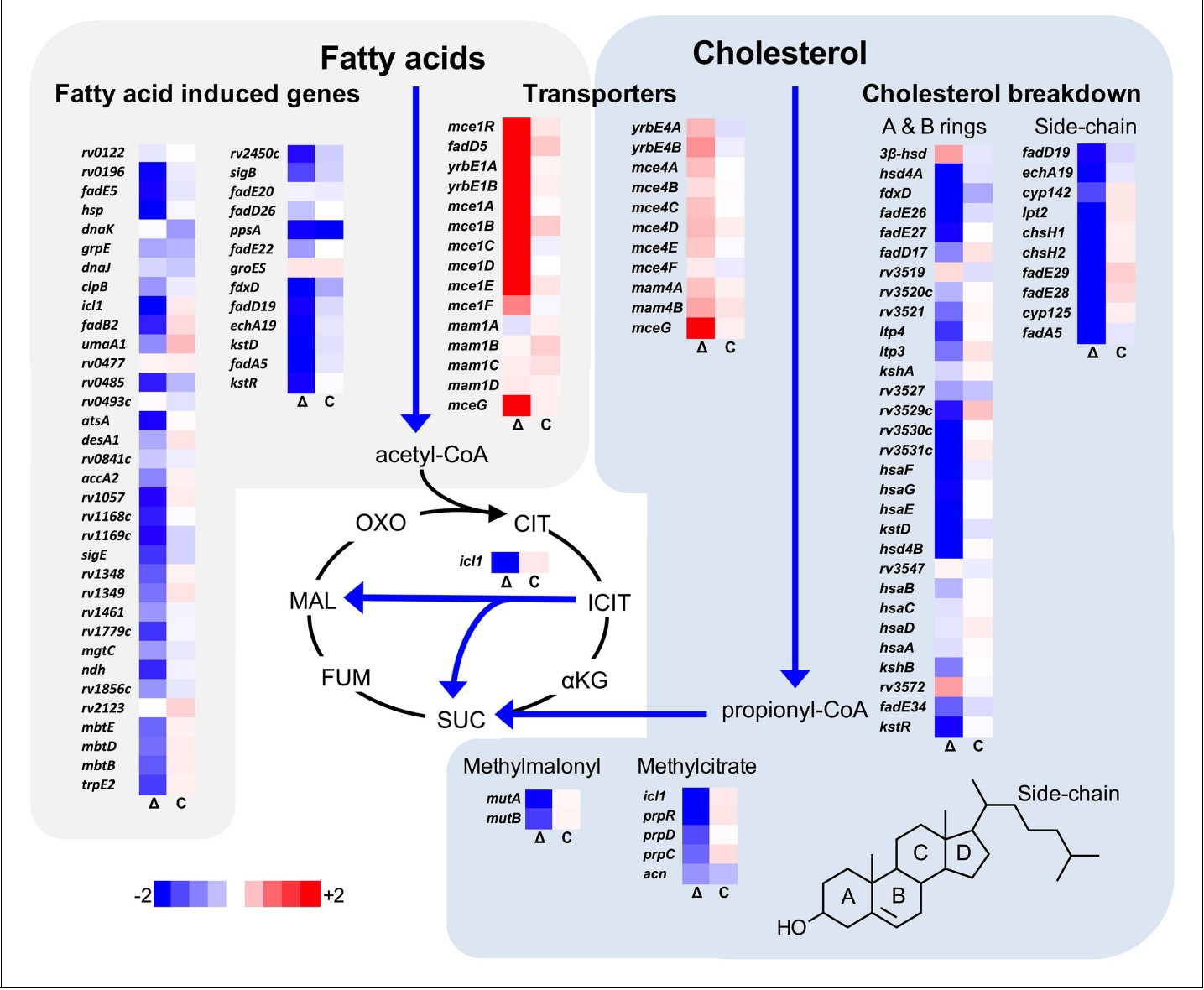

**Figure 2.** Transcriptional profile of the *ΔlucA*::hyg mutant during infection in macrophages. Bacterial gene expression profiles were determined at day 3-post infection in resting murine macrophages. Gene expression values from the *ΔlucA*::hyg mutant (*Δ*) and complement (C) strains were normalized to wild type. Data represent means (n = 3).

The following source data and figure supplement are available for figure 2:

**Source data 1.** Spreadsheet of curated gene expression data.

**Figure supplement 1.** Validation of cholesterol utilization defect with a GFP based reporter.

cholesterol (*Figure 2—figure supplement 1C*). We also quantified GFP fluorescence in the Mtb *ΔlucA*::hyg strain during infection in macrophages. To do this, intracellular bacteria were isolated from the infected macrophages and bacterial GFP fluorescence was quantified by flow cytometry. This analysis also demonstrated a ~ 75% reduction in GFP fluorescence in the Mtb *ΔlucA*::hyg strain (*Figure 2—figure supplement 1D*) supporting the idea that the Mtb *ΔlucA*::hyg mutant is defective in assimilating cholesterol during infection in macrophages.

## The Mtb *lucA*::hyg mutant does not assimilate fatty acids during macrophage infection

Unexpectedly, the transcriptional response of the Mtb *ΔlucA*::hyg mutant also revealed a gene expression signature that is consistent with a defect in fatty acid utilization. The relevant gene expression data is compiled in *Figure 2—source data 1*. For these analyses, we focused on the Mtb genes that are induced during infection in macrophages and during growth in axenic media containing palmitate (*Schnappinger et al., 2003*). The majority of these genes were strongly down-regulated in the *ΔlucA*::hyg mutant indicating a defect in fatty acid assimilation by the mutant (*Figure 2*).

To confirm that the *ΔlucA*::hyg mutant has a fatty acid utilization defect during infection in macrophages, we next quantified import of fluorescent palmitate (Bodipy-C16) by the intracellular bacteria. Resting murine macrophages were infected with wild type, *ΔlucA*::hyg mutant, and complemented strains, all of which constitutively express mCherry. On day 3, the infected macrophages were pulsed with Bodipy-C16 and confocal analysis revealed that the wild type and the complemented bacteria accumulated intracellular Bodipy-C16 as visible punctate inclusions, while the *ΔlucA*::hyg mutant did not (*Figure 3A*). To corroborate this finding, the bacteria were isolated from Bodipy-C16 pulse-labeled macrophages and we quantified the amount of Bodipy-C16 assimilation by flow-cytometery. This analysis revealed a 10-fold reduction in the amount of Bodipy-C16 assimilated by the Mtb *ΔlucA*::hyg mutant relative to the wild-type and complemented strains (*Figure 3B*). The decrease in Bodipy-C16 assimilation by the *ΔlucA*::hyg mutant is not due to a loss of bacterial viability since there was minimal difference in bacterial colony-forming units (CFU) at day 3 of these experiments (*Figure 3—figure supplement 1A*).

Mtb principally resides in an intracellular macrophage compartment that resembles an early endosome that fails to fuse with lysosomes (*Russell et al., 2010*). It is possible that the *ΔlucA*::hyg mutant is aberrantly trafficked to lysosomal compartments in the macrophage and this could potentially restrict access of Bodipy-C16 to the intracellular mutant. To rule this out, lysosomes were pulse labeled with Alexa647-dextran and at day 3 of infection we determined that wild type, the *ΔlucA*::hyg mutant, and complemented strain did not co-localize with the Alexa647-dextran-loaded lysosomes with Pearson coefficient of correlation values of 0.052 ± 0.062 (wild type), 0.101 ± 0.054 (*ΔlucA*::hyg), and 0.190 ± 0.174 (complemented strain) (*Figure 3—figure supplement 1B*).

During infection, Mtb can sequester and store fatty acids as triacylglycerol (TAG) within cytosolic intracellular lipid inclusions in the bacteria (*Daniel et al., 2011*). Therefore, the presence of lipid inclusions can serve as an indicator of fatty acid assimilation by Mtb during infection. To image intracellular lipid inclusions in Mtb, we used the neutral lipid stain Bodipy-493/503 (*Listenberger and Brown, 2007*). Staining infected murine macrophages with Bodipy-493/503 revealed a punctate staining pattern in the wild type and complemented strains while no staining was observed in the Mtb *ΔlucA*::hyg mutant (*Figure 3—figure supplement 1C*). The lack of Bodipy-493/503 staining in the *ΔlucA*::hyg mutant indicates that this strain does not accumulate fatty acids during infection in macrophages. Lipid inclusions in Mtb can also be visualized by transmission electron microscopy and we observed that only ~5% of the *ΔlucA*::hyg mutant cells had identifiable lipid inclusions, while 25% of the wild type and 35% of complemented bacteria had visible cytosolic lipid inclusions during infection in macrophages (*Figure 3C,D*).

The apparent decrease in cytosolic lipid inclusions within the *ΔlucA*::hyg mutant during infection in macrophages could also be due to enhanced bacterial turnover of the intracellular TAG in the mutant strain. To rule this possibility out, we treated the *ΔlucA*::hyg mutant bacteria with the broad-spectrum lipase inhibitor, tetrahydrolipstatin (THL) during infection in macrophages. It has been reported that THL can inhibit TAG turnover in Mtb (*Baek et al., 2011*). We predicted that if TAG were more efficiently degraded in the Mtb *ΔlucA*::hyg mutant, THL treatment would restore the presence of intracellular lipid inclusions in this mutant. Visualizing cytosolic bacterial lipid inclusions with Bodipy-493/503 following THL treatment revealed that THL had no effect on the presence of intracellular lipid inclusions in the *ΔlucA*::hyg mutant (*Figure 3—figure supplement 1D*). Together, these observations indicate that the Mtb *ΔlucA*::hyg mutant is unable to utilize both fatty acids and cholesterol during infection in macrophages.

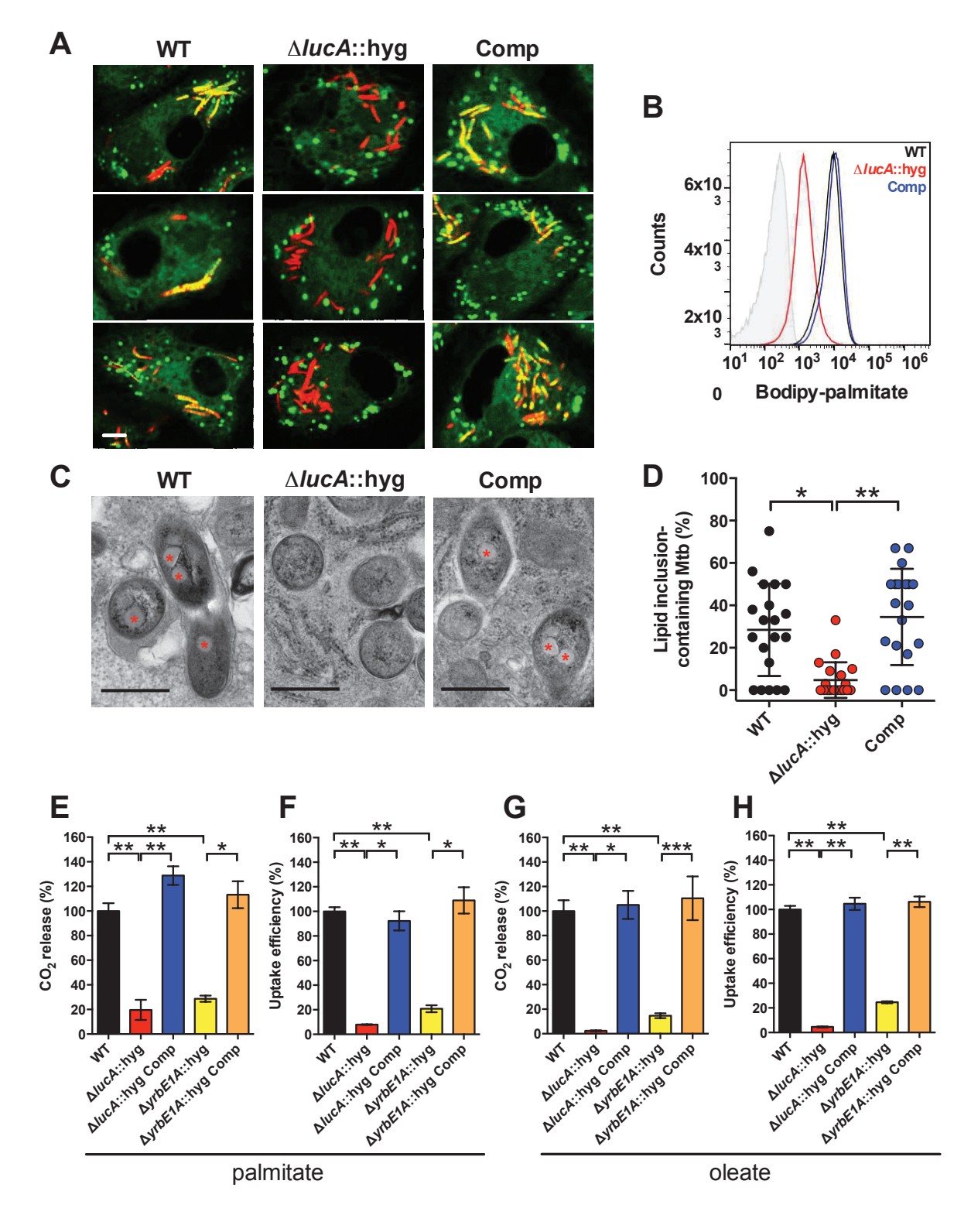

**Figure 3.** LucA facilitates fatty acid uptake during infection and in axenic culture. (**A**) Bodipy-C16 does not accumulate in Δ*lucA*::hyg mutant as cytosolic lipid inclusions. Representative confocal images of infected macrophages (red = mCherry Mtb, green = Bodipy-C16). Scale bar 5.0 μm. (**B**) Flow cytomeery quantification of Bodipy-C16 incorporation by Mtb isolated from pulse-labeled macrophages. Shaded histogram represents autofluorescence in the green channel. (**C**) Transmission electron microscopy reveals that lipid inclusions (indicated by asterisks) are not apparent in the

*Figure 3 continued on next page*

*Figure 3 continued*

Δ*lucA*::hyg mutant. Scale bar 0.5 μm. (**D**) Quantification of intracellular Mtb containing lipid inclusions per macrophage section. Horizontal bars are means ± SD (n = 20). (**E** and **G**) Catabolic release of $^{14}CO_2$ from [$^{14}$C(U)]-palmitic acid or [1-$^{14}$C]-oleic acid. (**F** and **H**) Uptake of [$^{14}$C(U)]-palmitic acid or [1-$^{14}$C]-oleic acid. The uptake rates were calculated from the incorporated radioactive counts (*Figure 3—figure supplement 2A,B*). Data are means ± SD (n ≥ 4). *p<0.0005, **p<0.0001, ***p<0.005 (Student's t test).

The following figure supplements are available for figure 3:

**Figure supplement 1.** Characterization of the mutant during infection in macrophages.

**Figure supplement 2.** Kinetic data used used in comparisons.

**Figure supplement 3.** Deletion of the full Mce1 operon leads to fatty acid uptake defect.

## LucA facilitates fatty acid uptake

Our studies using the macrophage infection model indicate that the Δ*lucA*::hyg mutant has a defect in fatty acid assimilation. Therefore, we next quantified fatty acid metabolism and uptake in the Mtb Δ*lucA*::hyg mutant using radiolabeled fatty acids in axenic culture. Fatty acid metabolism was measured by quantifying the catabolic release of $^{14}$C-$CO_2$ from [$^{14}$C(U)]-palmitate and [1-$^{14}$C]-oleate. This assay detected a ~80% and ~95% reduction in the Δ*lucA*::hyg mutant's ability to metabolize the [$^{14}$C(U)]-palmitate and [1-$^{14}$C]-oleate, respectively (*Figure 3E,G*). We used both of these lipid substrates because palmitate and oleate are the dominant fatty acid species found in mammalian cell membranes and we wanted to evaluate uptake of both saturated (palmitate) and unsaturated (oleate) substrates.

Fatty acid uptake was quantified with [$^{14}$C(U)]-palmitate and [1-$^{14}$C]-oleate similar to cholesterol uptake experiments. Relative to the wild-type and complemented strains, we observed a ~90% and ~95% reduction in the Δ*lucA*::hyg mutant's ability to assimilate [$^{14}$C(U)]-palmitate and [1-$^{14}$C]-oleate, respectively (*Figure 3F,H* and *Figure 3—figure supplement 2A,B*). Based on these results, we conclude that inactivating *lucA* perturbs fatty acid uptake by Mtb, which decreases the bacterium's ability to metabolize fatty acids.

## Mce1 is a fatty acid transporter

The system(s) responsible for fatty acid import in Mtb have remained elusive. While the Δ*lucA*::hyg mutant is defective in fatty acid uptake, the LucA protein lacks any recognizable domains that would predict a transport function for the protein. Thus, we hypothesized that, in the absence of *lucA*, Mtb may up-regulate expression of an actual fatty acid transporter to compensate for the fatty acid uptake defect observed in the mutant. Analyses of the up-regulated genes in the Δ*lucA*::hyg mutant during infection in macrophages revealed that genes in the *mce1* locus are strongly induced (*Figure 2*, *Figure 2—source data 1*). Thus, we hypothesized that Mce1 functions as a fatty acid transporter in Mtb. To test this hypothesis, we inactivated the Mce1 permease Rv0167/YrbE1A subunit by allelic exchange (Δ*yrbE1A*::hyg) and quantified fatty acid uptake and metabolism in this mutant. We found that the Δ*yrbE1A*::hyg mutant displayed a ~70% and ~85% reduction in its ability to metabolize [$^{14}$C(U)]-palmitate and [1-$^{14}$C]-oleate, respectively (*Figure 3E,G*). Relative to the wild-type and complemented strains, we detected a ~80% and ~75% reduction in the Δ*yrbE1A*::hyg mutant's ability to import [$^{14}$C(U)]-palmitate and [1-$^{14}$C]-oleate (*Figure 3F,H* and *Figure 3—figure supplement 2A,B*). Notably, Δ*yrbE1A*::hyg was still able to metabolize and import cholesterol to wild-type levels (*Figure 3—figure supplement 2C,D*). Lastly, we generated a separate mutant by inactivation of eight genes of the *mce1* locus by allelic exchange (Δ*mce1*::hyg). We found that the Δ*mce1*::hyg mutant was also defective in fatty acid uptake and metabolism, while this mutant had no detectable defect in cholesterol uptake or metabolism (*Figure 3—figure supplement 3*). Based on these results, we conclude that the Mce1 complex functions as a dedicated fatty acid transporter in Mtb.

## LucA interacts with Mce1- and Mce4-associated proteins

To shed light on the function of LucA, we next conducted a unbiased mycobacterial 2-hybrid screen to identify protein fragments that interact with LucA (*Singh et al., 2006*). For this, the $F_3$ domain of murine dihydrofolate reductase (mDHFR) was fused to the C-terminus of a full length LucA (LucA-$F_3$) (*Figure 4—figure supplement 1A*). We co-expressed LucA-$F_3$ in *M. smegmatis* (Msm) along with a library ($2 \times 10^6$) of random Mtb protein fragments fused to the $F_{1,2}$ domain of mDHFR (*Figure 4A*). Interactions between bait and prey fusions that carry the split domains of mDHFR confer resistance to trimethoprim in Msm. This protein interaction screen identified four sibling clones which all encode an in-frame N-terminal fragment (1–75 aa) of the Rv3492c/Mam4B protein fused to the $F_{1,2}$ domain of mDHFR. Mam4B is encoded by a gene within the *mce4* operon (*Figure 4A,G*) and this protein is predicted to function as a subunit of the Mce4 cholesterol transporter complex. To validate this screen result, we engineered a new truncated (TR) Mam4B fusion with the $F_{1,2}$ domain of mDHFR (Mam4B(TR)-$F_{1,2}$) (1–75 aa) and confirmed that co-expressing Mam4B(TR)-$F_{1,2}$ with LucA-$F_3$ in Msm conferred resistance to trimethoprim (*Figure 4B* and *Figure 4—figure supplement 1A*). We found that the transmembrane (TM) segment of Mam4B is the portion of the Mam4B protein that interacts with LucA. Co-expressing just the transmembrane domain (1–30 aa) of Mam4B fused to the $F_{1,2}$ domain of mDHFR (Mam4B(TM)-$F_{1,2}$) conferred trimethoprim resistance (*Figure 4B* and *Figure 4—figure supplement 1A*).

The Mtb genome encodes additional homologs of Rv3492c/Mam4B that are associated with the Mce1 transporter. Given that the *ΔlucA*::hyg mutant is also defective in the import of fatty acids, we hypothesized that LucA could also interact with homologous subunits of the Mce1 transporter. We focused on the homologous Mtb proteins Rv0175/Mam1A, Rv0177/Mam1C, Rv0178/Mam1D and Rv0199/OmamA because these proteins are predicted to be part of the Mce1 transporter (*Casali and Riley, 2007*; *Perkowski et al., 2016*) and conserve a similar transmembrane topology as Rv3492c/Mam4B (*Figure 4A*). To determine if these subunits also interact with LucA, we engineered $F_{1,2}$ fusions with various homologs of Rv3492c/Mam4B. We found that co-expressing LucA-$F_3$ with full length (FL) versions of Rv0177/Mam1C and Rv0199/OmamA fused to $F_{1,2}$ conferred trimethoprim resistance (*Figure 4B* and *Figure 4—figure supplement 1A*). Additionally, we found that LucA-$F_3$ with transmembrane domains (TM) of Rv0177/Mam1C and Rv0199/OmamA fused to $F_{1,2}$ conferred trimethoprim resistance (*Figure 4B* and *Figure 4—figure supplement 1A*). These data indicate that LucA physically interacts with subunits of the Mce1 and Mce4 transporters and suggests that the LucA protein participates in the function of these complexes.

## Substrate binding and translocation by Mce4 are dissociable events

Next we saught to determine if Rv3492c/Mam4B is required for cholesterol import. We inactivated Rv3492c/Mam4B by allelic exchange (*Δmam4B*::hyg) and quantified the rate of [4-$^{14}$C]-cholesterol uptake in this mutant. We found that relative to the wild type and complemented strains the *Δmam4B*::hyg mutant is still capable of importing [4-$^{14}$C]-cholesterol (*Figure 4C*). Importantly, we observed that the *Δmam4B*::hyg mutant has an obvious cholesterol metabolism defect resulting in an 80% reduction in the mutants ability to metabolize [4-$^{14}$C]-cholesterol (*Figure 4D*). The phenotype of the *Δmam4B*::hyg strain is specific to cholesterol given that the mutant has no defect in the uptake or metabolism of [1-$^{14}$C]-oleate (*Figure 4E,F*). Our interpretation of these observations is that the Rv3492c/Mam4B mutant binds and/or partly imports cholesterol across the Mtb cell envelope and this accounts for the uptake activity that we detect with this strain. Given that cholesterol is likely not metabolized by Mtb until the substrate reaches the bacterial cytosol we predict that the final translocation of cholesterol across the bacterial cytoplasmic membrane is defective in the *Δmam4B*::hyg mutant and a blockade at this step prevents metabolism of cholesterol in this strain. Together these data indicate that Rv3492c/Mam4B is required to complete the process of Mce4-mediated cholesterol internalization by Mtb.

## LucA stabilizes subunits of the Mce1 and Mce4 complexes

Recently, it was demonstrated that Rv0199/OmamA is involved in cholesterol metabolism in Msm and stabilizes the Mce1 complex in Mtb (*Perkowski et al., 2016*). Given that LucA interacts with Rv0199/OmamA and the related homologs Rv3492c/Mam4B and Rv0177/Mam1C, we hypothesized that both the Mce1 and Mce4 transporters may also be destabilized in the *ΔlucA*::hyg mutant. To

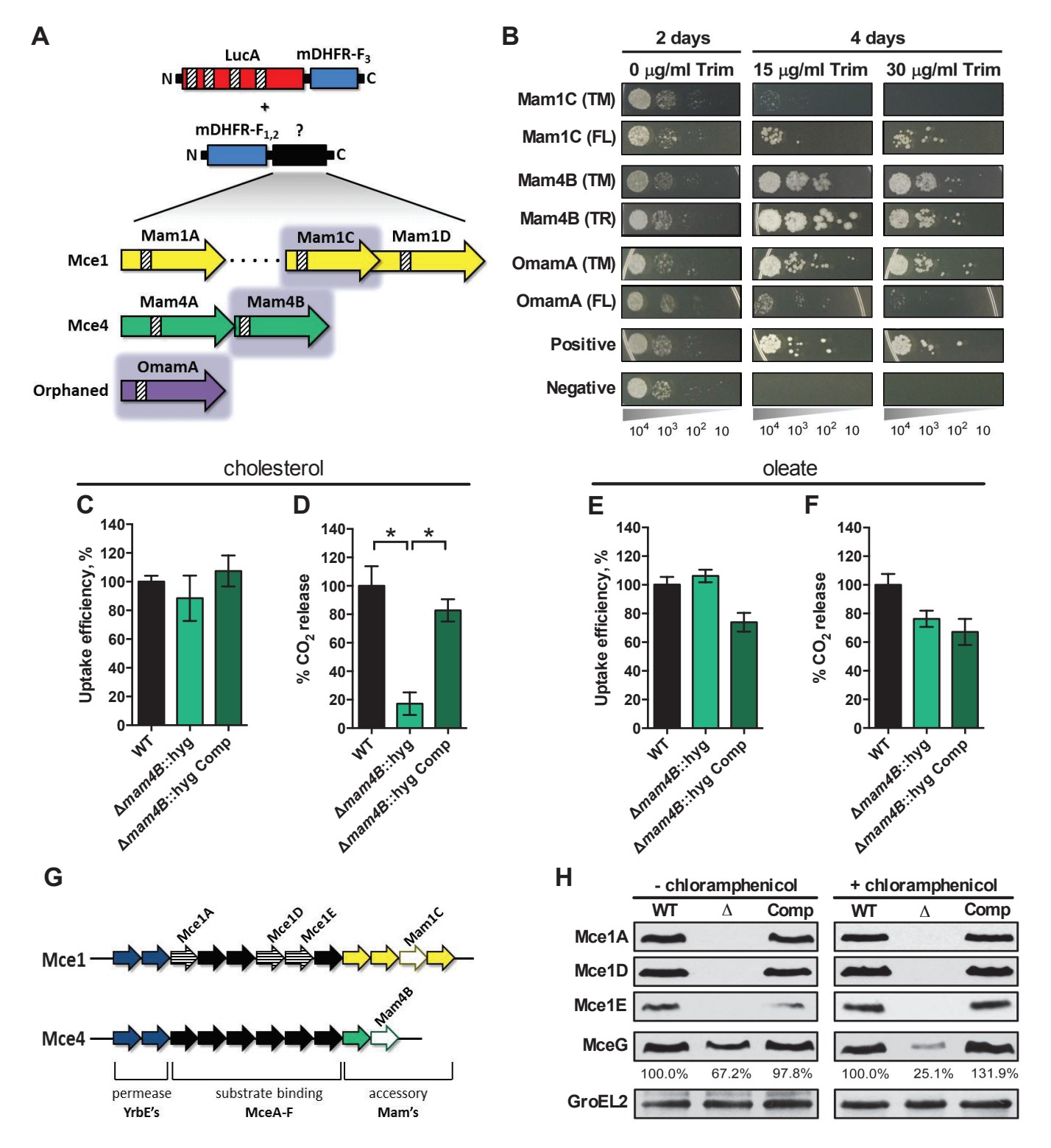

**Figure 4.** LucA interacts with subunits of Mce1 and Mce4 transporters and is required for their stability. (**A**) Schematic of split-DFHR protein constructs, shading indicates proteins that interact with LucA and striped boxes indicate predicted transmembrane domains. (**B**) Serial dilutions of Msm co-expressing the LucA-F$_3$ bait and the indicated prey were plated onto plates without antibiotic or plates containing trimethoprim at the indicated concentration. Growth on trimethoprim indicates protein-protein interaction. Positive is the positive control strain of Msm that co-expresses *Saccharomyces cerevisiae* homodimeric leucine zipper subunits (GCN4-F$_{1,2}$ and GCN4-F$_3$). Negative is the negative control strain of Msm that

*Figure 4 continued on next page*

*Figure 4 continued*

expresses LucA-F$_3$ and GCN4-F$_{1,2}$. (**C** and **D**) Cholesterol import and metabolism by the $\Delta mam4B$::hyg mutant. Data are means ± SD (n = 4).\*p<0.0001 (Student's t test). (**E** and **F**) Oleate import and metabolism by the $\Delta mam4B$::hyg mutant. Data are means ± SD (n = 4). (**G**) Organization of genes that encode the Mce1 and Mce4 transporters. Striped arrows indicate genes encoding proteins that are degraded in the $\Delta lucA$::hyg mutant. Empty arrows indicate genes encoding accessory proteins that interact with LucA. (**H**) Whole-cell lysates probed by western blotting using antibodies specific for the indicated proteins. Chloramphenicol was added for 2 days before the lysate was prepared where indicated. Inset values indicate MceG protein levels that were quantified and expressed as a ratio relative to MceG in the wild type lysates. GroEL2 is a loading control and blots are representative of two independent experiments.

The following figure supplement is available for figure 4:

**Figure supplement 1.** Details of interaction clones, gene expression analysis of Mce1 genes, and model of Mce mediated nutrient import.

test this, we generated whole-cell lysates from Mtb grown under the conditions used in the lipid uptake experiments. Gene expression analysis by qPCR confirmed that the *mce1* genes are expressed in the $\Delta lucA$::hyg mutant to equivalent levels relative to the wild-type and complemented strains (*Figure 4—figure supplement 1B*). In contrast, analysis of bacterial lysates revealed that, at the protein level, the putative subunits of the Mce1 complex (Mce1A, Mce1D, Mce1E) were completely degraded in the $\Delta lucA$::hyg mutant (*Figure 4G* and *Figure 4H*). Unfortunately, thus far we have been unable to raise antibodies specific to the analogous Mce4 subunits.

It is thought that MceG/Rv0655 functions as a common ATPase to hydrolyze ATP and facilitate nutrient uptake through the Mce transporters in Mtb (*Joshi et al., 2006*). MceG is required for optimum growth of Mtb on cholesterol and a mutant lacking MceG displays a cholesterol import defect that is equivalent to the cholesterol import defect observed in a mutant lacking Mce4 (*Pandey and Sassetti, 2008*). Although the role of MceG in fatty acid uptake has not been confirmed, it has been previously established that the stability of MceG requires co-expression of the Mce1 and Mce4 permease subunits in mycobacterial cells (*Joshi et al., 2006*). To further explore the cholesterol and fatty acid uptake defect in the $\Delta lucA$::hyg mutant, we hypothesized MceG may also be degraded or destabilized in this strain. Therefore, we quantified MceG levels in the bacterial lysates and observed a 30% decrease in amount of MceG in the whole-cell lysates of the $\Delta lucA$::hyg mutant (*Figure 4H*). It has recently been reported that synthesis of Mce proteins can exceed the rates of degradation (*Perkowski et al., 2016*) and given that there is negligible difference in the expression of MceG at the RNA level across the strains we used chloramphenicol to suppress protein synthesis in the bacteria. Under this condition, the relative amount of MceG protein decreased by 75% in the $\Delta lucA$::hyg mutant (*Figure 4H*). These results demonstrate that, in the absence of LucA, not only subunits of the Mce1 transporter complex, but also shared MceG ATPase, are degraded, possibly due to activity of unknown specific proteases. These data provide an explanation for the linked defect in both fatty acid and cholesterol uptake observed in the $\Delta lucA$::hyg mutant and that LucA serves to coordinate the uptake of these nutrients by protecting the Mce1 and Mce4 transporters from degradation (*Figure 4—figure supplement 1C*).

## LucA contributes to the in vivo fitness of Mtb

These data indicate that LucA is involved in the utilization of two critical host-derived lipid substrates. It is known that deletion of one or both of the Mce1 and Mce4 transporters leads to decreased survival of the pathogen in mouse model infection (*Joshi et al., 2006*), and we predict that both these transporters would be nonfunctional in the $\Delta lucA$::hyg mutant. In human macrophages, the $\Delta lucA$::hyg mutant displayed a growth lag that culminates in a 10-fold difference in bacterial counts compared to wild-type and complemented strains over a 7-day infection period (*Figure 5A*). A similar phenotype was observed in the resting murine macrophages where the $\Delta lucA$::hyg mutant replicated poorly over a 10-day infection period (*Figure 5B*). In both human and murine macrophages, the final CFU counts for the $\Delta lucA$::hyg mutant remained close to initial inoculum levels. These data are consistent with the hypothesis that Mtb requires LucA to sustain maximal growth on cholesterol and/or fatty acid substrates in macrophages.

In the lung tissues of C57BL/6J mice, the $\Delta lucA$::hyg mutant also demonstrated a fitness defect and did not attain levels of bacterial burden comparable to either the wild type or complemented

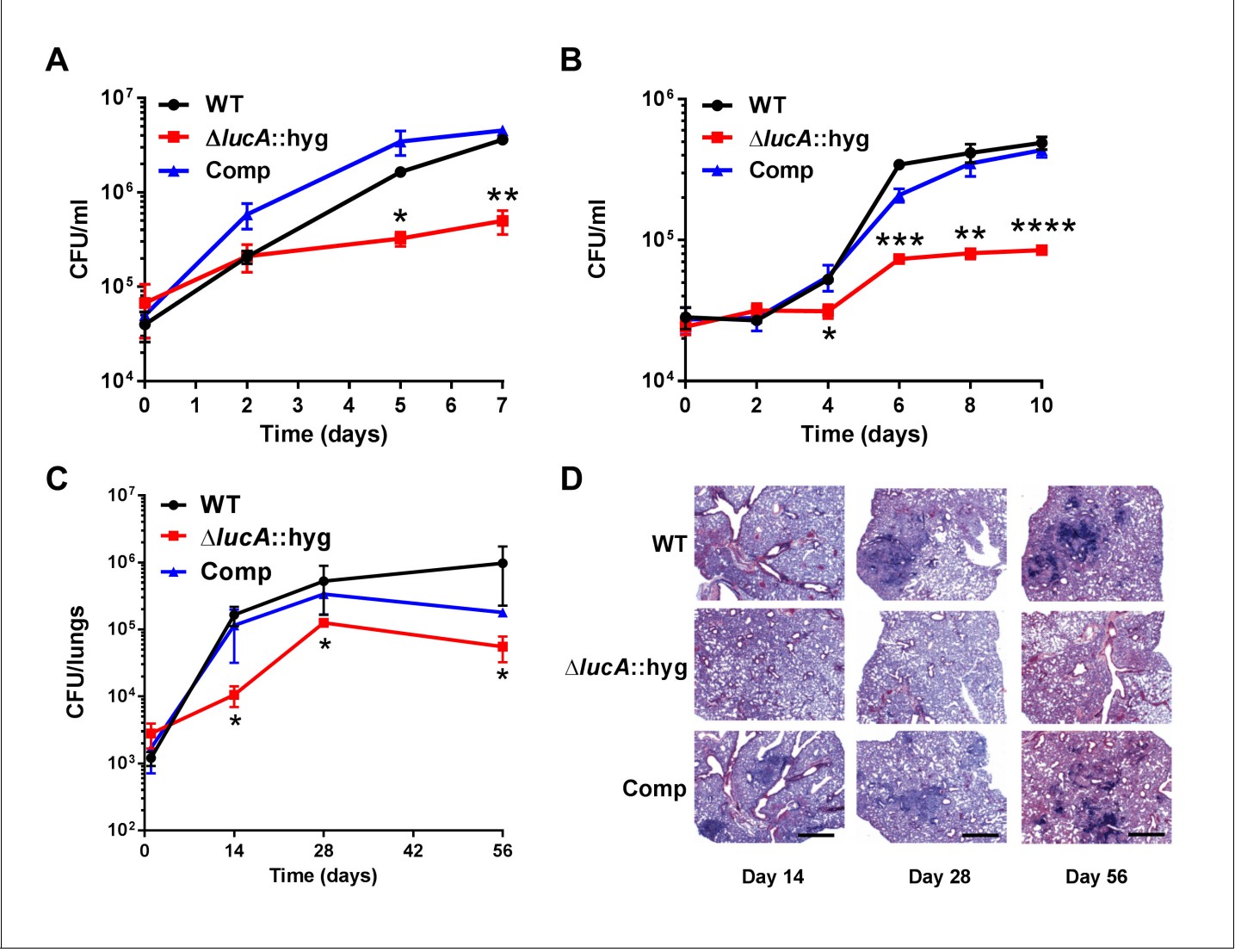

**Figure 5.** LucA is required for the full fitness of Mtb in macrophages and survival in mouse lungs. (**A**) Bacterial replication in resting human monocyte derived macrophages. Data are means ± SD (n = 3). (**B**) Bacterial replication in resting murine bone-marrow-derived macrophages. Data are means ± SD (n = 3). (**C**) Bacterial survival in mouse lung tissues. Data are means ± SD (n = 5 per time point) (**D**) Lung pathology of infected mice collected at indicated time points and H&E stained. Scale bar 400 μm. *p<0.05, **p<0.005, ***p<0.001, ****p<0.0005 (Student's t test).

strains. At day 14 post-infection, 10-fold fewer CFU's of the Δ*lucA*::hyg mutant were recovered relative to wild-type and the complemented strains. During the remainder of the 56-day infection period, growth of the Δ*lucA*::hyg mutant remained restricted, leading to three- to fivefold reduction in viable Δ*lucA*::hyg bacteria (*Figure 5C*). The Δ*lucA*::hyg mutant also induced less pulmonary pathology throughout the entire course of infection (*Figure 5D*). Complementation almost completely restored pathogenicity and lung tissue pathology confirming the specificity of the mutant phenotype to the *lucA* gene.

## Discussion

While there is general agreement as to the role of Mce4 in cholesterol uptake in Mtb, the process of fatty acid assimilation by the bacterium has remained enigmatic. These data shed new light on the coordination of fatty acid and cholesterol import and reveal that a network of proteins associates with the Mce1 and Mce4 transporters to integrate the uptake of both fatty acids and cholesterol.

Using an unbiased forward genetic screen, we discovered that transposon insertions in the *lucA* gene rescue cholesterol toxicity in an Mtb strain that lacks *Δicl1*. Subsequent analysis confirmed that mutation of *lucA* (*ΔlucA*::hyg) has a profound impact on cholesterol uptake (*Figure 1*) and that this cholesterol uptake defect confers growth rescue in the *Δicl1 lucA*::TnHimar double mutant.

Transcriptional profiling revealed that genes normally activated by cholesterol are down-regulated in the *ΔlucA*::hyg mutant during infection in macrophages (*Figure 2*). This is consistent with a defect in cholesterol uptake in this mutant. We were surprised to discover that the *ΔlucA*::hyg mutant also produces a gene expression signature indicative of a fatty acid metabolism defect. Although the functions for many of these 'fatty acid-induced' genes are unknown, their expression pattern serves as a reliable indicator of a fatty acid uptake defect in Mtb. The majority of the 20 most highly expressed genes in the *ΔlucA*::hyg during infection in macrophages map to the *mce1* locus. Up-regulation of genes encoding the Mce1 fatty acid transporter in the *ΔlucA*::hyg mutant during infection likely reflects an attempt by the bacterium to compensate for the absence of fatty acids that normally fuels Mtb's metabolism. The mechanism that controls expression of the *mce1* locus in response to fatty acid depletion is unknown but we hypothesize that Mtb has an ability to sense metabolite pools to control synthesis of the Mce1 fatty acid transporter complex.

The Mtb cell envelope constitutes formidable barrier to the transport of any hydrophobic molecule. Actinomycetal bacteria with mycolic-acid-containing cell walls that are capable of metabolizing cholesterol use the Mce4 complex to import the sterol (*Mohn et al., 2008*; *Pandey and Sassetti, 2008*; *Perkowski et al., 2016*) and this has led to the idea that all four of the Mce complexes transport hydrophobic molecules across the Mtb cell wall. Various studies have linked Mce1 to Mtb virulence (*Gioffré et al., 2005*; *Joshi et al., 2006*; *Shimono et al., 2003*) but the function of the Mce1 complex was hitherto unknown. It has been reported that inactivating Mce1 in Mtb induces a lipid homeostasis defect and the accumulation of free mycolic acids in the Mtb cell wall (*Cantrell et al., 2013*; *Forrellad et al., 2014*). Based on these observations, it was hypothesized that the Mce1 may transport fatty acids and/or mycolic acids across the cell wall/membrane of Mtb but it was reported that Mce1 mutant in Mtb displayed a minor defect in fatty acid uptake (*Forrellad et al., 2014*). We used assays comparable to those previously described and detected major perturbations in fatty acid assimilation in our Mce1 mutants (*Figure 3*). We cannot fully explain the discrepancy between these findings, but we have noticed that spontaneous mutants of Mtb unable to produce phthiocerol dimycocerosate (PDIM) assimilate less fatty acid compared to PDIM positive strains. Our work was conducted in a PDIM-positive strain of Mtb Erdman, and we think this has allowed us to detect Mce1-mediated fatty acid transport.

Studies with *Mycobacterium leprae* (Mlep) are consistent with the interpretation that Mce1 functions as a mycobacterial fatty acid transporter. The Mlep genome contains a single *mce* locus which is most similar to the *mce1* operon found in Mtb and Mlep is fully capable of importing and metabolizing palmitate when the bacteria are recovered from animal tissues (*Franzblau, 1988*). The Mlep genome also encodes homolog of *lucA* (ml2337) suggesting that LucA could be central for Mce1 function in this bacterium. Given that the Mlep genome does not conserve genes that encode proteins for cholesterol import/metabolism and this bacterium does not metabolize cholesterol (*Marques et al., 2015*), it is likely that ml2337 has one role in Mlep. The finding that LucA facilitates fatty acid uptake and stabilizes components of the Mce1 transporter provides additional evidence that Mce1 functions as a fatty acid transporter. Therefore, the fatty acid uptake defect in the *ΔlucA*::hyg mutant is most likely a consequence of the degradation of Mce1 components (*Figure 4*).

The *mce1-4* loci make up four separate operons in the Mtb genome and each operon encodes the putative protein subunits that likely comprise the individual Mce transporters. It is thought that each Mce transporter is substrate-specific and the individual subunits are predicted to perform discrete roles. Specifically, the *mce4* operon encodes two putative permease subunits (YrbE4A and YrbE4B), six cell-wall-associated Mce proteins (Mce4A-Mce4F), and two accessory subunits (Mam4A and Mam4B). We found that LucA interacts with Mam4B and that Mam4B is required for the metabolism of cholesterol; however, the mutant lacking Mam4B is still capable of binding cholesterol. Based on this observation, we propose that Mce4 imports cholesterol via a two-step process that involves cholesterol binding/shuttling across the cell wall followed by the final translocation through the cytoplasmic membrane delivering cholesterol into the cytosol. The accessory and permease subunits may participate in the final translocation of cholesterol across the cytoplasmic membrane while the Mce proteins likely participate in the binding/shuttling of cholesterol across the Mtb cell

envelope. Binding/shuttling of lipids across the Mtb cell envelope may be analogous to what has been proposed for lipid trafficking by Mce proteins across the periplasm of gram-negative bacteria (*Malinverni and Silhavy, 2009*; *Nakayama and Zhang-Akiyama, 2017*; *Thong et al., 2016*). This two-step mechanism of nutrient uptake may be a generalizable mechanism for all Mce transporters. Additional support for the two-step model comes from our observation that the Mce1 proteins (Mce1A, Mce1D, and Mce1E) are degraded in the *ΔlucA*::hyg mutant and this strain is unable to bind/shuttle fatty acids across the Mtb cell envelope. Lastly, we can detect a residual level of fatty acid and cholesterol uptake and metabolism in mutants lacking Mce1 and Mce4, respectively. It is very likely that compensatory systems function to import these lipids in the absence of Mce1 and Mce4 and we are currently testing this hypothesis.

Recently, it was reported that orphaned Mce associated protein Rv0199/OmamA facilitates cholesterol utilization in Msm and stabilizes the Mce1 complex in Mtb (*Perkowski et al., 2016*). Similarly, we found that LucA is also required to stabilize subunits of the Mce1 transporter. This work demonstrates that LucA interacts with the Mce1 and Mce4 accessory subunits Rv0199/OmamA, Rv0177/Mam1C and Rv3492/Mam4A. Based on this, we predict that LucA is recruited to the Mce1 and Mce4 complexes to stabilize or assemble the transporters via interactions with the accessory subunits. Given that MceG is required for cholesterol uptake and is thought to facilitate the import of additional nutrients in Mtb (*García-Fernández et al., 2017*; *Joshi et al., 2006*; *Pandey and Sassetti, 2008*), our finding that the MceG protein is also destabilized in the *ΔlucA*::hyg may explain the various nutrient uptake defects in this mutant. Homology searches based on three-dimensional structures identified a putative protease inhibitor domain within the N-terminus of LucA (*Kelley et al., 2015*). We hypothesize that LucA may locally inactivate a protease to maintain the integrity of the transporter complexes. Regulating activity of the transport complexes through proteolysis could be a mechanism to rapidly halt nutrient uptake through the Mce transporters, however such a suggestion will require more research.

During growth in the presence of cholesterol Mtb shunts cholesterol-derived methylmalonyl-CoA (originating from propionyl-CoA) toward the increased synthesis of methyl-branched, cell wall polyketide lipids (*Griffin et al., 2012*; *Jain et al., 2007*; *Yang et al., 2009*). This metabolic shunting requires that sufficient fatty-acid-derived acyl-AMP primers are available to support biosynthesis of polyketide lipids (*Quadri, 2014*). When excess fatty acids are supplied to infected macrophages, Mtb can enhance the flux of propionyl-CoA into polyketide lipids such as PDIM during infection (*Lee et al., 2013*). It would be advantageous for Mtb to regulate fatty acid assimilation to maintain the acyl-AMP pools required for efficient synthesis of methyl-branched lipids. Thus, coordination of cholesterol and fatty acid uptake by LucA could ensure that balanced levels of these nutrients are maintained for optimized metabolism.

The central carbon and lipid metabolic pathways of Mtb have emerged as potential drug targets (*Rhee et al., 2011*; *VanderVen et al., 2015*); therefore, understanding the bottlenecks or weaknesses in these pathways will assist TB drug discovery. Additionally, the flux of fatty acids into TAG and central metabolism contributes to drug tolerance in Mtb (*Baek et al., 2011*), a phenotype that is further enhanced by immune pressure during in vivo infection (*Liu et al., 2016*). Targeting the specialized lipid metabolic pathways in Mtb that are involved in fatty acid and cholesterol utilization could be a viable strategy for the development of new drugs that reduce Mtb drug tolerance and augment current TB drug regimens. Our data have defined new participants in the complex processes of fatty acid and cholesterol assimilation by Mtb. A better understanding of the functional integration of Mtb's specialized metabolic pathways is required to acquire a fuller appreciation of Mtb pathogenesis.

## Materials and methods

### Bacteria and growth conditions

*M. tuberculosis* strains were routinely grown at 37°C in 7H9 (broth) or 7H11 (agar) media supplemented with OAD enrichment (oleate-albumin-dextrose-NaCl), 0.05% glycerol and 0.05% tyloxapol (broth). AD enrichment consisted of fatty acid free albumin-dextrose-NaCl. 7H9-based minimal medium is composed of Difco Middlebrook 7H9 powder 4.7 g/l, 100 mM 2-(*N*-morpholino)ethanesulfonic acid pH 6.6, and carbon sources as indicated. Cholesterol was added to the liquid and solid

media as tyloaxapol:ethanol micelles as described (*Lee et al., 2013*). Hygromycin 100 μg/ml, kanamycin 25 μg/ml, streptomycin 50 μg/ml, and apramycin 50 μg/ml were used for selection. For *E. coli* selection hygromycin was used at 150 μg/ml.

### Transposon screen and strain construction

A library of transposon mutants ($\sim10^5$) in a $\Delta icl1$-deficient strain of Mtb described by (*Lee et al., 2013*) was plated onto 7H11 OAD agar containing 100 μM cholesterol. Individual mutants were recovered in culture. Chromosomal DNA was isolated and the transposon insertion sites were PCR amplified and sequenced according to *Prod'hom et al. (1998)*. Mutant strains of Mtb were generated by allelic exchange (*Mann et al., 2011*) using a hygromycin resistance cassette mutant. Allelic exchange was confirmed by sequencing and/or Southern analysis using the Direct nucleic acid labeling and detection kit, GE Health Care, Marlborogh, MA. All the strains used in the study are summarized (*Supplementary file 1*).

### Radiorespirometry assays

Lipid oxidation was monitored by quantifying the release of $^{14}CO_2$ from [4-$^{14}$C]-cholesterol, [$^{14}$C(U)]-palmitate, and [1-$^{14}$C]-oleate by radiorespirometry as described (*VanderVen et al., 2015*). Briefly, Mtb cultures were pre-grown in 7H9 AD for 5 days. Then they were incubated at $OD_{600}$ of 0.7 in 5 ml 7H9 AD spent medium supplemented with 1.0 μCi of radiolabeled substrates in vented standing T-25 tissue culture flasks placed in a sealed air-tight vessel with an open vial containing 0.5 ml 1.0 M NaOH at 37°C. After 5 hr, the NaOH vial was recovered, neutralized with 0.5 ml 1.0 M HCl, and the amount of base soluble $Na_2^{14}CO_3$ was quantified by scintillation counting. The radioactive signal was normalized to the relative levels of bacterial growth by determining the $OD_{600}$ for the bacterial cultures. % CO2 release was expressed as a ratio of normalized radioactive signal for each strain relative to the wild-type control.

### Lipid uptake assays

Lipid uptake was quantified as described previously (*Forrellad et al., 2014*; *Pandey and Sassetti, 2008*) with slight modifications. Briefly, Mtb was cultured at an initial $OD_{600}$ of 0.1 in 7H9 AD medium in vented standing T-75 tissue culture flasks. After 5 days, cultures were normalized to $OD_{600}$ of 0.7 in 8 ml using spent medium, and 0.2 μCi of radiolabeled substrates was added to bacteria. After 5, 30, 60, and 120 min of incubation at 37°C 1.5 ml of the bacterial cultures were collected by centrifugation. Each bacterial pellet was washed thrice in 1 ml of ice-cold wash buffer (0.1% Fatty acid free-BSA and 0.1% Triton X-100 in PBS), fixed in 0.2 ml of 4% PFA for 1 hr. The total amount of radioactive label associated with the fixed pellet was quantified by scintillation counting. The radioactive signal was normalized to the relative levels of bacterial growth, that is, to the $OD_{600}$ of the bacterial cultures before addition of radioactive label. The uptake rate was calculated by applying linear regression to the normalized radioactive counts over time, and uptake efficiency was expressed as a ratio of uptake rate for each strain relative to the wild type control.

### Macrophage isolation and culturing

Macrophages were differentiated using bone marrow cells from BALB/c mice Jackson Laboratories, Bar Harbor, ME and maintained in DMEM supplemented with 10% heat inactivated fetal calf serum, 2.0 mM L-glutamine, 1.0 mM sodium pyruvate, 10% L-cell-conditioned media and antibiotics (100 U/ml penicillin and 100 mg/ml streptomycin) at 37°C and 7.0% $CO_2$ for 10 days before infection. Human macrophages were differentiated from purified human peripheral blood mononuclear cells obtained from Elutriation Core Facility, University of Nebraska Medical Center and maintained in DMEM supplemented with 10% pooled heat inactivated human serum SeraCare, Milford, MA, 2.0 mM L-glutamine, 1.0 mM sodium pyruvate and antibiotics (100 U/ml penicillin and 100 mg/ml streptomycin) at 37°C and 7.0% $CO_2$ for 7 days before infection. Media without antibiotics were used for infections with Mtb.

### Transcriptional profiling

Murine bone-marrow-derived macrophages were seeded into two T-75 tissue culture flasks (1.5 × $10^7$ cells per flask) and infected with Mtb at a MOI of 4:1 for 3 days. Bacterial RNA was isolated,

amplified, dye labeled, and hybridized to the microarray as described (*Liu et al., 2016*; *Rohde et al., 2007*). Gene expression data have been deposited in the NCBI Gene Expression Omnibus database accession number (GSE98792) (*Edgar et al., 2002*).

### *prpD'*::GFP reporter assays

The promoter of *rv1130/prpD* was fused to GFP in a replicating vector that constitutively expresses mCherry (*Supplementary file 1*) To detect prpD promoter activity bacteria were grown in 7H9-based minimal medium containing 10 mM glucose for 5 days, washed twice with PBS 0.05% tyloxapol, and passed to medium containing 100 µM cholesterol or propionate at the indicated concentration for 24 hr. The bacteria were fixed with 4% paraformaldehyde (PFA) and GFP expression was quantified by flow cytometery on a BD Biosciences *LSR II* flow cytometer. To detect prpD promoter activity during infection bacteria grown in 7H9-based minimal medium containing 10 mM glucose were used to infect murine macrophages at an MOI of 5:1. After 24 hr infection, the macrophages were fixed with 4% PFA, scraped into 10 ml of PBS and suspended in 1 ml of lysis buffer (0.1% SDS, 0.1 mg/ml Proteinase K in $H_2O$). Macrophages were lysed by 25 passages through a 25-gauge needle, the bacteria containing cell lysate was centrifuged, the pellet was retained and analyzed on a BD Biosciences *LSR II* flow cytometer. Flow cytometry data were analyzed using FlowJo (Tree Star, Inc).

### Imaging of intracellular lipid inclusions

Confluent monolayers of macrophages in Ibidi eight-well glass-bottom chambers were infected with bacteria at a MOI of 4:1. Extracellular bacteria were removed after 4 hr of infection. Infected macrophages were maintained in cell culture medium at 37°C and 7% $CO_2$ for 3 days. Lipid inclusions of bacteria in macrophages were metabolically labeled with Bodipy-C16 (final concentration 20 µM) conjugated to 1.0% de-fatted bovine serum albumin (BSA) for a 30-min pulse followed by a 1-hr chase with fresh media. Live-cell images were acquired as described (*Podinovskaia et al., 2013*). For lipid staining, macrophages were transferred onto sterile coverslips in 24-well plates, infected with Mtb for 3 days and fixed in 4% PFA followed by staining with BODIPY-493/503 (1.0 µg/ml, at room temperature for 1 hr). Post-acquisition, images were analyzed using Volocity (PerkinElmer Life Sciences).

### Flow cytometric quantification of assimilated lipids

Murine bone-marrow-derived macrophages were seeded into T-150 tissue culture flasks ($3 \times 10^7$ cells per flask) and infected with Mtb at a MOI of 4:1. After 3 days of infection, Bodipy-palmitate (final concentration 8 µM) conjugated to de-fatted 1% BSA was added to the cells for 1-hr pulse and then chased with cell media for another hour. The infected macrophages were scraped into 15 ml of homogenization buffer (250 mM sucrose, 0.5 mM EGTA, 20 mM HEPES,. 05% gelatin, pH 7.0) and pelleted by centrifugation at 514xG (1500 rpm, Beckman Allegra 6KR centrifuge, GH-3.8 rotor), followed by cell lysis by 70 passages through a 25-gauge needle. 5 ml of cell lysate was centrifuged at 146xG (800 rpm) for 10 min, supernatant (suspensions of phagosomes) was retained and treated with 0.1% Tween-80 at 4°C for 15 min to break-open Mtb containing vacuoles. Isolated bacteria were washed once in PBS + 0.05% tyloxapol and fixed in 4% PFA. Flow cytometry data were collected on BD FACS LSR II and analyzed using FlowJo (Tree Star, Inc). Flow cytometric quantification analyses are described in more detail at Bio-protocol (*Nazarova et al., 2018*).

### Colocalization studies with Alexa647-dextran-labeled lysosomes

At day 3 of infection, bone-marrow-derived macrophages were pulse labeled with 50 µg/ml Alexa647-dextran for 45 min and chased in fresh media for an additional 45 min. Following the chase period, the infected cells were fixed and imaged by confocal microscopy. An extended focus merge of the two channels and the background was used to threshold the data as described (*Costes et al., 2004*) and colocalization was calculated using Volocity (PerkinElmer Life Sciences).

### Transmission electron microscopy

Imaging was conducted as described (*Podinovskaia et al., 2013*).

## Tetrahydrolipstatin treatment of infected macrophages

100 µM tetrahydrolipstatin Sigma, St Louis, MO was added to infected bone-marrow-derived macrophages at 4 hr post infection and maintained in the medium throughout the infection. Cells were fixed at day 3, stained with Bodipy-493/503 Invitrogen Carlsbad, CA, and imaged by confocal microscopy.

## Protein fragment complementation screen

Library construction and the screen were performed as described (*Singh et al., 2006*). Briefly, Mtb genomic DNA was isolated and partially digested with AciI and HpaII, size fractionated (0.5–2 kb), and cloned into the ClaI site of pUAB300 upstream of the $F_{1,2}$ domain of murine dihydrofolate reductase. *E. coli* MegaX DH10B T1 Life Technologies Carlsbad, CA electrocompetent cells were used for transformation. In total $5 \times 10^5$ independent clones were selected on LB hygromycin agar plates. The clone library was isolated by QIAGEN QIAfilter Plasmid Giga Kit and used to transform Msm mc$^2$155 containing pUAB200 which co-expresses the bait protein LucA fused to the $F_3$ (LucA-$F_3$) domain of murine dihydrofolate reductase. In total $2 \times 10^6$ clones were screened on plates containing trimethoprim 30 µg/ml. Clones containing fragments of the Mtb dihydrofolate reductase (*rv2763c/dfrA*) were identified by PCR and removed (85.8%). Inserts from the *dfrA*-negative clones were sequenced, and the only in-frame clone that was identified more than once (four times) contained first 225 bp of *rv3492c/mam4B*.

## Spotting two-hybrid interaction assay

All Msm clones expressing the bait and prey constructs used for interaction assays are shown (*Figure 4—figure supplement 1A*). Msm was grown in modified 7H9-OAD media containing 2% glycerol, 0.5% additional glucose and 0.05% Tween-80 shaking at 37°C. ~$8.3 \times 10^6$ bacteria/ml culture was grown for 5 hr before serial dilutions were made to spot onto agar plates containing trimethoprim at 0, 15, and 30 µg/mL.

## Quantification of protein interactions

To quantify the strength of interactions ($IC_{50}$) we used an AlamarBlue based approach modified from *Singh et al. (2006)*. Briefly, log-phase cultures of Msm clones were transferred into 96-well microtiter plates at a density of $10^6$ of cells per well. Eight 2-fold serial dilutions of trimethoprim were made for each clone, from 600 to 4.69 µg/ml. The final volume in the wells was 200 µl. After 41 hr of incubation at 37°C, 30 µl 50% AlamarBlue Life Technologies, Carlsbad, CA (diluted with the media) was added to the wells and after incubating for 20 hr the fluorescence intensity was measured in Gemini EM Microplate Reader (Molecular Devices) with excitation at 530 nm and emission at 590 nm. 100% inhibition was assigned to the wells without bacterial cells, and 0% inhibition to the wells with cells without trimethoprim.

## Antibody generation and western analysis

Antibody for MceG was generated in rabbits using the peptide KAQAAILDDL conjugated to keyhole limpet hemocyanin by Cocalico Biologicals, Stevens, PA. This peptide was used for antibody purification by immunoaffinity chromatography. To generate the lysate, bacteria were grown as for the lipid uptake assays. In the cases of chloramphenicol treatment, the antibiotic was added at 20 µg/ml 2 days prior to harvesting the bacteria. To generate lysates bacteria were washed twice with PBS 0.05% tyloxapol and fixed with 4% PFA for 1 hr. Fixed cells were washed twice in PBS 0.05% tyloxapol and lysed by sonication. Protein concentrations were determined by BCA Thermo Fisher Scientific, Waltham, MA and equivalent amounts of protein were resolved by SDS-PAGE and transferred to nitrocellulose membranes. The primary anti-Mce1A, anti-Mce1D and, anti-Mce1E antibodies were obtained from Christopher Sassetti (*Feltcher et al., 2015*), and the anti-GroEL *antibody was obtained from BEI resources*. A HRP-conjugated goat-anti rabbit IgG Jackson ImmunoResearch, West Grove, PA was used as the secondary antibody. ImageJ was used to quantify signals on Western blots.

## qPCR

Bacteria were cultured as described for western analysis and the RNA was extracted and analyzed as previously described (*Abramovitch et al., 2011*).

## Bacterial survival assay in macrophages

Confluent macrophage (human and murine) monolayers in 24-well dishes were infected with Mtb at a MOI 4:1 for murine cells and a MOI of 0.5:1 for the human cells. Extracellular bacteria were removed by washing with fresh media after 4 hr of infection. At indicated time points, macrophages were lysed with 0.1% Tween-80 in water and the lysates were serially diluted in 0.05% Tween-80 in water. The lysates were plated on 7H11 OAD agar and CFU were quantified after 3–4 weeks incubation at 37°C.

## Mouse infections

Eight-week-old female C57BL/6J WT mice Jackson Laboratories were infected with 1000 CFU of Mtb Erdman (wild type, *ΔlucA*::hyg, complemented strain) via an intranasal delivery method as described (*Sukumar et al., 2014*). This was accomplished by lightly anesthetizing the mice with isoflurane and administering the bacteria in a 25 μl volume onto both nares. At sacrifice, the lungs were removed and half of the lungs were fixed in 4% PFA overnight, while another half was used for bacterial load quantification. For the latter, lungs were homogenized in PBS 0.05% Tween-80 and plated on 7H11 OAD agar. CFU were quantified after 3–4 weeks incubation at 37°C.

## Lung histopathology

PFA fixed lung lobes were stained with hematoxylin and eosin by the Cornell Histology Laboratory. Stained sections were imaged using a Zeiss Axio Imager M1 equipped with an AxioCam Hrc camera.

## Acknowledgements

We thank Linda Bennett for excellent technical support, Robert Abramovitch, Shumin Tan, John Helmann, and Lu Huang for productive discussions, Maria Podinovskaia for assistance with imaging, Yancheng Liu for help with library construction for M-PFC. We also thank Adrie Steyn for the generous gift of the M-PFC vectors, Martin Pavelka for the aacC4 apramycin resistance cassette, and Christopher Sassetti for the Mce1 antibodies. This work was supported by the NIH grants (AI099569 and AI119122) to BCV and (AI118582 and AI067027) to DGR.

## Additional information

### Funding

| Funder | Grant reference number | Author |
| --- | --- | --- |
| National Institute of Allergy and Infectious Diseases | AI118582 | David G Russell |
| National Institute of Allergy and Infectious Diseases | AI067027 | David G Russell |
| National Institute of Allergy and Infectious Diseases | AI099569 | Brian C VanderVen |
| National Institute of Allergy and Infectious Diseases | AI119122 | Brian C VanderVen |

The funders had no role in study design, data collection and interpretation, or the decision to submit the work for publication.

### Author contributions

EVN, Conceptualization, Investigation, Visualization, Methodology, Writing—original draft, Writing—review and editing; CRM, TL, KMW, NS, WL, SC, Investigation; DGR, Funding acquisition, Writing—review and editing; BCV, Conceptualization, Formal analysis, Funding acquisition, Investigation, Visualization, Methodology, Writing—original draft, Writing—review and editing

## Author ORCIDs

Brian C VanderVen, http://orcid.org/0000-0003-3655-4390

## Ethics

Animal experimentation: All animal care and experimental protocols were in accordance with the NIH "Guide for the Care and Use of the laboratory Animals" and were approved by the Institutional Animal Care and Use Committee of Cornell University (protocol number 2013-0030).

## Additional files

### Supplementary files

• Supplementary file 1. Details of strains.

### Major datasets

The following dataset was generated:

| Author(s) | Year | Dataset title | Dataset URL | Database, license, and accessibility information |
|---|---|---|---|---|
| Nazarova EV, Montague CR, Russell DG, VanderVen BC | 2017 | Transcriptional response of Mycobacterium tuberculosis ΔlucA::hyg and complemented strains vs. WT in resting murine bone marrow-derived macrophages | https://www.ncbi.nlm.nih.gov/geo/query/acc.cgi?acc=GSE98792 | Publicly available at the NCBI Gene Expression Omnibus (accession no: GSE98792) |

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
