## [Decision Letter]

Thank you for submitting your article "Rv3723/LucA coordinates fatty acid and cholesterol uptake in *Mycobacterium tuberculosis*" for consideration by *eLife*. Your article has been favorably evaluated by Gisela Storz (Senior Editor) and three reviewers, one of whom, Bavesh D Kana (Reviewer #1), is a member of our Board of Reviewing Editors. The following individuals involved in review of your submission have agreed to reveal their identity: Heran Darwin (Reviewer #2) and William Jacobs (Reviewer #3).

The reviewers have discussed the reviews with one another and the Reviewing Editor has drafted this decision to help you prepare a revised submission.

Summary:

In their submission, Nazarova and colleagues investigate cholesterol metabolism and transport in *Mycobacterium tuberculosis*. Using a forward genetic screen, with an Icl1 deficient mutant of *M. tuberculosis* that is unable to grow on cholesterol, they identify genes for cholesterol utilization in media containing a mixture of fatty acids and simple carbohydrates. The premise of this gain of function assay is that secondary mutations in the Icl1 deficient background would rescue the growth defect by alleviating the accumulation of toxic metabolites. This analysis yielded 19 candidate genes, 16 were further confirmed using growth curve assays. Of the 16, 9 genes had already been previously identified as important for growth on cholesterol as the sole carbon source. Some of these directly alleviate methylcitrate toxicity and others reduce the flux of propionyl-CoA. One of the candidates identified was lucA, which has been implicated in cholesterol utilization in *M. tuberculosis*, but the mechanism remained unknown. The authors confirm that lucA is membrane associated and not cytoplasmic and further proceed to construct a mutant of *M. tuberculosis* that is defective for lucA. As expected, the lucA mutant does not metabolize cholesterol, an effect due to reduced uptake of cholesterol. The authors then determine the transcriptional profile of the lucA mutant when infected in mouse bone marrow-derived macrophages and find reduced expression of the kstR regulon, which they attributed to reduced cholesterol uptake. To further confirm this, the authors construct a reporter to monitor the activity of the prpD promoter, which responds to the presence of cholesterol. The authors also identify a fatty acid utilization defect for the mutant when infected in macrophages. They confirm this defect using fluorescently labelled palmitate and Bodipy-493/503, the latter staining for triacylglycerides (TAGs). They also confirm that reduced TAG accumulation in the lucA mutant was not due to enhanced turnover of intracellular lipids. Using assays with labeled metabolites, the authors demonstrate that fatty acid uptake is reduced in the lucA mutant. As LucA lacks any structural resemblance to transporters, the authors further study the above mentioned defects by generating a mutant defective for Mce1, a putative transporter, and through this confirm that the Mce1 complex is responsible for fatty acid, but not cholesterol, transport. Using bacterial two-hybrid screens, the authors show that LucA interacts with subunits of the Mce1 and Mce4 transporters. Furthermore, this interaction seems to stabilize the Mce1 complex. Also levels of MceG – an ATPase required for transport – were reduced in the lucA mutant. The authors next demonstrate that a mutant defective for Mce4 could take up cholesterol but could not utilize it. Finally, the authors demonstrate that LucA is important for successful colonization of macrophages and is also required for successful infection in the murine model of tuberculosis infection. This an excellent tour de force that utilizes a clever forward genetic screen to identify mutants of *M. tuberculosis* defective in cholesterol utilization in the presence of fatty acids. Overall, the work identifies and characterizes a new player in an important metabolic pathway (lipid metabolism) and the authors provide a convincing and compelling set of arguments that validate their conclusion.

Essential revisions:

1) The manuscript is incredibly dense, with a diversity of methodological approaches and modalities of representing data. As it stands, it is not suitable for the broad readership base of *eLife* and requires substantive streamlining. There were many supplementary figures, the relevance of which was not always clear. Please reflect on how to integrate the supplementary material into the logic of the manuscript in a more careful and considered way. The last figure with mouse data should just be incorporated in Figure 1 where lucA is first described. It fits better here as it obfuscates the manuscript when placed later on.

2) Related to the above, the authors need to strongly reassert their discovery of a previously unknown function of Rv3723. It needs to be clear that this work provides a rational basis for redefining Rv3723 a lipid utilization coordinator (LucA). Thus this needs to be changed in the Abstract. Furthermore, the definition of the gene should occur in the Results and not in the Introduction, and should be reiterated in the Discussion.

---

## [Author Response]

*Essential revisions:*

*1) The manuscript is incredibly dense, with a diversity of methodological approaches and modalities of representing data. As it stands, it is not suitable for the broad readership base of eLife and requires substantive streamlining. There were many supplementary figures, the relevance of which was not always clear. Please reflect on how to integrate the supplementary material into the logic of the manuscript in a more careful and considered way. The last figure with mouse data should just be incorporated in Figure 1 where lucA is first described. It fits better here as it obfuscates the manuscript when placed later on.*

*2) Related to the above, the authors need to strongly reassert their discovery of a previously unknown function of Rv3723. It needs to be clear that this work provides a rational basis for redefining Rv3723 a lipid utilization coordinator (LucA). Thus this needs to be changed in the Abstract. Furthermore, the definition of the gene should occur in the Results and not in the Introduction, and should be reiterated in the Discussion.*

The original submission was compressed to adhere to the *eLife* word limit guidelines and we agree it was quite dense. For this submission we have added additional explanation around the rationale for the experiments. We have made the data more accessible to a broad readership and we have expanded our descriptions throughout.

Many different approaches are used in this research and some methods are described here for the first time. We described these methods in more detail and have streamlined the presentation of the data. In the Introduction we expanded on the role of MCE proteins in other bacteria and plants to help put our work in the context of other biological systems.

We carefully organized the supplementary data and linked the most relevant supplementary data to the main figures. The previous submission had 26 figures or supplements. This version has 13 figures and supplements that present the data in a more streamlined fashion.

We largely kept the original sequence of the data and kept the mouse data at the end. This way we can tell the story beginning with the screen all the way through determining the function of Rv3723 LucA in a single block. We believe that presenting mouse data after the screen disrupts the flow of the story. Going from a screen to finding an uncharacterized gene to a mouse study is awkward to justify and we feel that a broad readership is likely to be more interested in the mechanism/function of Rv3723/LucA while the TB research community will mostly care about a phenotype in animals.

In this submission we added some new data. The previous submission did not present complementation data for the *Δmam4A::*hyg or the *Δmce1* mutants in our biological assays. Although we were not asked to include this we added this data in Figure 4 and a supplement for Figure 3 as additional bars in the graphs. We made a measured but stronger assertion that the function of Rv3723 was previously unknown in the abstract and throughout. We moved the definition of the gene from the Abstract to the Results.